# Intra- and Interspecies RNA-Seq Based Variants in the Lactation Process of Ruminants

**DOI:** 10.3390/ani12243592

**Published:** 2022-12-19

**Authors:** Mohammad Farhadian, Seyed Abbas Rafat, Christopher Mayack, Mehdi Bohlouli

**Affiliations:** 1Department of Animal Science, Faculty of Agriculture, University of Tabriz, Tabriz 51666-16471, Iran; 2Molecular Biology, Genetics, and Bioengineering, Faculty of Engineering and Natural Sciences, Sabancı University, Istanbul 34956, Turkey; 3Institute of Animal Breeding and Genetics, Justus-Liebig-University Giessen, 35390 Giessen, Germany

**Keywords:** cow, milk, RNA-Seq, sheep, variant calling

## Abstract

**Simple Summary:**

RNA-Seq data provide a new chance to find transcriptome variants. We used RNA-Seq data to detect the variants involved in the three different stages (before peak = BP, peak = P, and after peak = AP) of the lactation process in two sheep and two cow breeds. Furthermore, several KEGG pathways and enriched gene ontologies associated with immune system activation and the metabolic process were demonstrated by analyzing the functional enrichment of the genes that were affected. Findings of the present study also suggest that milk yield and milk composition in sheep and cow breeds at different stages of lactation can be related to known and novel variants of specific genes related to milk fat and protein synthesis. The results pave the way for further studies on determining the genetic basis of milk production. The novel variants discovered here using RNA-seq data may be central and crucial when it comes time to design new SNP chips used as guides for selective breeding programs.

**Abstract:**

The RNA-Seq data provides new opportunities for the detection of transcriptome variants’ single nucleotide polymorphisms (SNPs) in various species and tissues. Herein, milk samples from two sheep breeds and two cow breeds were utilized to characterize the genetic variation in the coding regions in three stages (before-peak (BP), peak (P), and after-peak (AP)) of the lactation process. In sheep breeds Assaf and Churra, 100,462 and 97,768, 65,996 and 62,161, and 78,656 and 39,245 variants were observed for BP, P, and AP lactation stages, respectively. The number of specific variants was 59,798 and 76,419, 11,483 and 49,210, and 104,033 and 320,817 in cow breeds Jersy and Kashmiri, respectively, for BP, P, and AP stages. Via the transcriptome analysis of variation in regions containing QTL for fat, protein percentages, and milk yield, we detected a number of pathways and genes harboring mutations that could influence milk production attributes. Many SNPs detected here can be regarded as appropriate markers for custom SNP arrays or genotyping platforms to conduct association analyses among commercial populations. The results of this study offer new insights into milk production genetic mechanisms in cow and sheep breeds, which can contribute to designing suitable breeding systems for optimal milk production.

## 1. Introduction

The process of milk secretion from mammary glands is referred to as lactation. As a dynamic and multifaceted biological process, lactation is a pivotal part of the reproduction system [1]. In a majority of mammalian species, the amount of milk production follows a curved pattern over the course of lactation. In early lactation, milk production peaks following an initial rise. After the peak yield, production gradually decreases until the end of lactation [2]. The ability of an animal to continue producing milk at a high level after the peak yield is referred to as lactation persistency [3]. Improvement in lactation persistency can increase total milk production when milk yield and lactation persistency are correlated [2].

Thorough knowledge of lactation biology at the molecular level facilitates the possibility of identifying genes and single-nucleotide polymorphisms (SNPs) that are associated with milk production traits (e.g., milk yield, protein percentage, protein yield, fat percentage, and fat yield) in livestock breeding programs [4]. Variation in milk compositions in different lactation stages can be determined by assessing transcriptional regulation and detecting SNPs in underlying genes associated with the desired traits [2]. Various regulatory and metabolic pathways producing fatty acids, carbohydrates, and amino acids are also included in the lactation process, which can determine the milk’s nutritional quality [2]. In dairy cattle, for example, whey and casein protein genes are differently expressed in different lactation stages. During such alterations, transcriptionally-regulated genes are reported to have receptor activity, signal transducer activity, and enriched catalytic activity [5].

To date, diverse genomic approaches such as gene expression analysis and genome-wide association studies (GWAS) have been proposed to describe the possible genetic background for milk yield and composition traits in different breeds of sheep and cattle species. Following a meta-analysis of the RNA-Seq dataset, Farhadian et al. (2020) reported that the genes *GJA1*, *FBXW9*, *AP2A2*, *NPAS3*, *CDKN2C*, *HOXC9*, *INTS1*, and *SFI1* as potential candidates associated with milk-related attributes [6]. Moreover, the significance of cell proliferation, fat metabolism, milk protein production, cell differentiation, and immune competency in the lactation process is stressed [6]. In sheep, polymorphisms in principal milk proteins (whey and caseins) have been assessed in several studies to detect possible associations [7]. The transcriptomic approach has been adopted to identify the genetic variants expressed in the mammary glands of lactating sheep. Such an approach leads to identifying numerous pathways and genes harboring mutations influencing dairy production traits [8]. Using GWAS, Pedrosa et al. [9] explored the associated variants with lactation persistency, milk yield, fat yield, fat percentage, protein yield, and protein percentage in North American Holstein cattle.

Since most sheep and cow production traits are complex, extensive studies have also focused on dairy quantitative trait loci (QTL) mapping. Nevertheless, the traditional methodology adopted for QTL mapping with low/middle-density SNP genotyping platforms or genome-wide sparse microsatellite markers complicates the detection of the actual causal mutations that underlie these multifaceted attributes [8].

High-throughput RNA sequencing analysis is performed for gene expression profiling [6]. The benefits of RNA-Seq for effective SNP detection in transcribed genes have been reported in various species and tissues [10,11]. In comparison to DNA sequencing, SNP calling analysis relies on RNA-Seq data and is a more cost-effective approach, with an almost 100% reliable rate [11]. Moreover, the majority of SNPs detected through SNP calling analysis are placed in the transcribed regions of the genome, in which variants are most likely to result in phenotypic variations and endure selection pressure [12].

Thus, this study aimed to adapt RNA-Seq for identifying gene-based SNPs in two cow and sheep breeds and to examine the association of identified SNPs with the variation in milk production trait, across different lactation stages. Besides the overall characterization of variabilities in the cow and sheep milk transcriptome, this analysis focused on the recognition of variations in the coding regions that contain QTL for milk composition and yield characteristics, important milk enzymes, and milk fat metabolism-related proteins.

## 2. Materials and Methods

### 2.1. Data Collection

In the present study, we utilized a lactation process-related milk transcriptome dataset obtained from the European Nucleotide Archive (ENA) database; the dataset was comprised of RNA-Seq information from two *Bos taurus* and *Ovis aries* species (Table 1).

For both species, specimens were available for three different lactation stages (i.e., before peak, BP; peak, P; and after peak, AP). The first dataset (SRP125676) contained mammary epithelial cells from Kashmiri and Jersey dairy cattle breeds. In these cattle breeds, the number of days in milk production: 15, 90, and 250, were considered as BP, P, and AP, respectively. For BP and AP, the SRP125676 dataset contained three specimens per cattle breed. For P, two and three specimens were available for Kashmiri and Jersey cattle breeds, respectively. The second dataset (SRP065967) contained milk somatic cells from Assaf and Churra dairy sheep breeds. In sheep breeds, the number of days in milk production: 10 and 50 days, were considered as BP and P, respectively, and the number of days in milk production, 120 and 150, were assigned as AP.

### 2.2. Processing the RNA-Seq Data

The raw read qualities of the RNA-Seq data were evaluated in FastQC 0.11.9 [14]. The low-quality reads were trimmed with Trimmomatic 0.32 [15] using the following parameters: TRAILING: 3, SLIDINGWINDOW: 4:20, MINLEN: 40. We aligned the cleaned and trimmed reads of cow and sheep, respectively, on the *Ovis aries* (Oar_v4.0) and *Bos taurus* (ARS-UCD1.2) genomes, using Hisat2 (0.1.5-beta) [16].

### 2.3. SNPs Calling, Alignment, and Annotation

Following the alignment, we marked the duplicate reads via the MarkDuplicates tool from Picard 2.8.1 (https://picard.sourceforge.net/ (25 August 2014) software, and these were excluded in the following steps. Subsequently, the GATK 4.0 [17] pipeline (i.e., including “Base Recalibration,” “Split’N’Trim,” and “HaplotypeCaller”) was applied for variant calling from the aligned reads. A primary list of the identified variants was filtered out based on standard quality metrics (HomopolymerRun > 5, total depth of coverage < 10, RMSMappingQuality < 40, QualitybyDepth < 2, MappingQualityRankSum < −12.5, and ReadPosRankSum < −8). We compared the identified SNPs for two species aiming to describe SNP associations with milk yield and composition across three lactation stages. The breed-stage-specific SNPs were utilized for downstream analysis. We performed the Ensembl’s Variant Effect Predictor tool (VEP, v97.0) [18] to identify and annotate the SNP location. The impact of the identified SNPs, along with their positions in the genes, was characterized with the VEP tool. Furthermore, the sorting intolerant from tolerant (SIFT) algorithm of the VEP tool was used to identify the effect of missense single amino acid substitution. Moreover, SIFT was used as a sequence homology-based algorithm that predicts whether an SNP is tolerated (SIFT score ≤ 0.05) on the basis of the degree of evolutionary conservation between homologous proteins in multiple species [19].

### 2.4. QTL Analysis

In order to identify SNP associations with the production of milk attributes genetically, we performed a co-localization analysis with milk-associated QTLs. All the milk production-related cow and sheep QTLs were attained from AnimalQTLdb (Release 44) [20]. This version of the sheep QTLdb included 3572 QTLs denoting 273 traits, and this version of cow QTLdb was comprised of 161,781 QTLs, expressing 680 traits. Among sheep QTLs, 130 QTLs were related to five traits associated with milk composition. Milk yield characteristics that were considered included: “Milk yield,” “Milk protein percentage,” “Milk protein yield,” “Milk fat percentage,” and “Milk fat yield”. Of the cow QTLs, 2101 QTLs were related to five milk production attributes, and these were taken into account as: “Milk yield,” “Milk protein percentage,” “Milk protein yield,” “Milk fat percentage,” and “Milk fat yield”. Next, we compared breed-stage-specific SNP positions for every breed with the locations of QTLs.

### 2.5. Functional Enrichment Analysis

The genes harboring breed-stage-specific SNPs were utilized for KEGG pathway and gene ontology (GO) analysis to detect the enriched biological process associated with milk composition and milk yield properties. For this purpose, the enrichplot package was used [21]. We considered the false discovery rate correction (FDR < 0.01) as the cut-off threshold of statistical significance for identifying the significant KEGG and GO terms.

The graphical abstract of different bioinformatics analysis steps for SNP calling in two sheep breeds and two cow breeds is presented in Figure 1.

## 3. Results

### 3.1. Mapping

A total of four datasets, which contain two species and two breeds, and the related lactation process, were selected. Our goal was to identify SNPs across three different stages of lactation, namely BP, P, and AP. Finally, including 47 samples, were selected for variant calling. The samples were divided into BP, P, and AP stages to identify SNPs. Each lactation period included 3/2/3 and 3/3/3 samples for Jersy and Kashmiri cow breeds for BP, P, and AP, respectively. Moreover, Assaf and Churra breeds contained 4/4/7 and 4/4/7 samples in each BP, P, and AP period, respectively. More information on samples and mapping results can be seen in Appendix A.

### 3.2. Variant Calling and Functional Annotation

For BP, P, and AP stages of lactation, the GATK pipeline resulted in 100,462/97,768/65,996 SNPs in the Assaf sheep breed and 62,161/78,656/39,245 SNPs in the Churra sheep breed, respectively. Of these, 78,645 (78.2%)/77,175 (78.9%)/51,938 (78.8%) and 48,775 (78.5%)/61,639 (78.3%)/30,286 (77.1%) SNPs were annotated as known SNPs in the Ensembl ovine SNP database in Assaf and Churra sheep breeds for BP, P, and AP stages of lactation respectively and were considered for further analysis (Appendix A). The number of identified variants in Jersy and Kashmiri cow breeds for BP, P, and AP stages were 59,798/76,419/11,483 and 49,210/104,033/320,817, respectively. Of these, 50,955 (85.2%)/68,349 (89.5%)/99,067 (88.9%) and 43,140 (87.7%)/92,880 (89.3%)/299,171 (93.3%) SNPs were annotated as known SNPs in the Ensembl bovine SNP database in Jersy and Kashmiri cow breeds for BP, P, and AP stages of lactation, respectively and were considered for further analysis (Appendix A). The percentage and number of known and novel SNPs in Ensemble ovine and the bovine SNP database are presented in Table 2, and the known SNPs were used for further analysis.

For the BP stage, 32,078 SNPs overlapped between two sheep breeds, and 17,867 SNPs overlapped between Jersy and Kashmiri cattle breeds. At the P stage of lactation, 34,121 and 28,058 common SNPs were found in sheep and cow breeds, respectively. Also, there were 7598 and 41,430 common SNPs at the AP stage of lactation in sheep and cow breeds, respectively.

Investigation of Assaf and Churra sheep breeds showed 24,089/25,292/1150 and 11,564/25,138/2333 breed-stage-specific SNPs for BP, P, and AP stages, respectively (Figure 2A,B; Appendix A). Also, in the Jersy and Kashmiri cow breeds, there were 12,468/14,862/47,103 and 5785/13,903/215,074 breed-stage-specific SNPs for BP, P, and AP stages, respectively (Figure 3A,B; Appendix A).

Functional prediction results of the breed-stage-specific SNPs are summarized in Table 3. The results show that the impact pattern of SNPs was similar for all breeds and stages except in the BP and P stages of the Jersy breed. High-impact SNPs were much less frequent than modifier, moderate, and low-impact SNPs. Moreover, a similar pattern of SNP locations on the genome was found for all breeds and stages.

Two different patterns of SNP locations on the genome were found for three different stages of lactation in four breeds (Table 3). There were 4908/3494/141 and 2560/5036/544 specific exonic SNPs in Assaf, and Churra breeds in the BP, P, and AP stage, 1329/911/46 and 677/1313/140 were missense (non-synonymous) SNPs, 196/131/4 and 97/158/21 of which were predicted as damaging variants (or deleterious SNPs). Annotation analysis in the Jersy and Kashmiri breeds showed that there were 2811/2113/2403 and 1761/2509/13,729 specific exonic SNPs at BP, P and AP stage 1024/709/826 and 536/905/4261 were missense, 160/131/134 and 105/173/529 were deleterious SNPs (Table 3).

By locating Assaf and Churra specific-SNPs of BP, P, and AP stages in milk-related QTL regions, 4710/8834/1892 and 3421/7537/714 SNPs were found, respectively. Among the BP, P, and AP stages specific-SNPs of Jersy and Kashmiri breeds in milk-related QTL regions, 1325/1673/63,780 and 672/4277/179,275 SNPs, respectively, were detected in QTL position ranges (Table 4).

### 3.3. Variants in Milk Protein Related Genes

Variability associated with milk protein was investigated in the genes coding for major milk proteins; some of these include encoding caseins [casein α-S1 (*CSN1S1*), casein α-S2 (*CSN1S2*), casein β (*CSN2*), and casein κ (*CSN3*)], whey proteins [α-lactalbumin (*LALBA*), and β lactoglobulin (*PAEP*)] [22]. After variant filtration in Assaf and Churra breeds in the BP, P, and AP stages of lactation, a total of 320/388/91 and 101/179/97 variants were identified within these genes. Moreover, in the Jersy and Kashmiri breeds in the BP, P, and AP stages 63/40/94 and 86/77/167 variants were detected within these genes. Among these variants in Assaf and Churra in the BP, P, and AP stages of lactation, 96/107/0 and 8/68/44 variants were found to be novel, and 224/281/91 and 93/111/53 variants were previously annotated in SNPdb (version v97.0). In addition, in Jersy and Kashmiri breeds in the BP, P, and AP stages of lactation, 15/4/13 and 14/11/22 variants were found to be novel, and 48/36/81 and 72/66/145 variants were previously annotated in SNPdb (version v97.0) (Table 5).

A high number of the variants found in the genes coding for major milk proteins were positioned within introns. Among all variants in milk protein genes, 48 were missense variants, out of which 42 and 6 were tolerate and deleterious variants, respectively. Among the missense variants detected in this study, ten were in *PAEP*, eight were in *LALBA*, 11 were in *CSN2*, 13 were in *CSN3*, five were in *CSN1S1*, and one was in *CSN1S2* (Appendix A).

In the *PAEP* gene, our analysis identified two missense variants, namely rs430610497 and rs109625649. The rs430610497 variant was in all three lactation stages of the sheep breeds, and rs109625649 was in all three stages of lactation in the Kashmiri breed and the BP stage of the Jersy cow breed. The rs430610497 and rs109625649 variants substitute Histidine/Tyrosine and Alanine/Valine, respectively.

Three known SNPs, rs403176291, rs465119286, and rs722550244, were detected within the *LALBA* gene. All of these SNPs are missense variants corresponding to the alteration of Alanine/Valine, Isoleucine/Valine, and Arginine/Glutamine substitutions, which are classified as tolerated variants. The rs403176291 variant was identified for sheep breeds in the BP and P lactation stages. In addition, the rs465119286 variant, was detected in the three stages of lactation in the Kashmiri breed. The last variant, rs722550244 was detected in the AP stage of the Kashmiri breed.

Regarding the *CSN2* gene, three SNPs, rs43703013, rs43703011, and rs109299401, were found in cow breeds only, and all of them were tolerate and substitute SNPs that included Arginine/Serine, Histidine/Proline, and Methionine/Leucine alterations, respectively. Out of which, the rs43703013 and rs43703011 variants were identified in all three stages of lactation in the Kashmiri breed. On the other hand, the rs109299401 SNP was only detected in the BP stage of the Jersy breed and the two others. The rs43703013 and rs43703011 were identified in the BP and AP stages within the Jersy breed.

Regarding the *CSN3* gene, our analysis identified three missense variants namely, rs43703016, rs43703015, and rs450402006. Out of which, one SNP, rs43703016, was found in all three stages of lactation in both cow breeds, and this was a deleterious and substitute SNP an Alanine/Aspartic acid alteration. Furthermore, the rs43703015 SNP was detected in all three stages of lactation in both the Jersy and Kashmiri breeds and was a tolerate and substitute SNP, that caused an Isoleucine/Threonine alteration. The last SNP, rs450402006, was only identified in the AP stage, in the Kashmiri breed, and was a substitute SNP that included a Threonine/Isoleucine alteration.

One already described missense SNP, rs43703010, was detected within the *CSN1S1* gene in all three stages of the Kashmiri breed and the AP stage of the Jersy breed; this was considered to be a tolerate alteration. This mutation causes the following amino acid change: glutamic acid/glycine.

Two tolerate and missense SNPs, rs465436451 and rs476152522, found in the AP stages of the two cow breeds only, were considered to be substitute alterations and include Threonine/Alanine and Valine/Phenylalanine changes in the *CSN1S2* gene, respectively.

Six novel missense variants were detected in milk protein genes (Table 6). Out of the SNPs detected in this study, two were in *LALBA*, three were in *CSN1S2*, and one SNP was within the *PAEP* gene. Of these, four belong to the Jersy breed and two belong to the Kashmiri breed.

Based on Table 6, all novel variants were identified in cow breeds and within the P and AP stages of lactation. Among the novel variants, four were within the AP stage, and two were within the P stage of lactation. Furthermore, one of them, which was in the *LALBA* gene, was categorized as a deleterious mutation.

### 3.4. Variants in Milk Fat-Related Genes

The milk-fat related genes can be clustered following lipid metabolism processes: fatty acid synthesis and desaturation (*ACACA*, *FASN*, and *SCD*), lipid droplet formation (*BTN1A1*, *XDH*), fatty acid activation and intracellular transport (*ACSL1*, *ACSS2*, and *FABP3*), acetate triacylglycerol synthesis (*GPAM*), fatty acid import cells (*LPL* and *VLDLR*), and other genes related with lipids metabolism like *PLIN2* [23]. To find SNPs in the genes related to milk fat content, we filtered the mutations located within a total of 12 genes that have been previously related to milk fat metabolism [13].

After variant filtration in Assaf and Churra in the BP, P, and AP stages of the lactation, a total of 790/864/518 and 722/504/369 variants were identified within these genes. In addition, in Jersy and Kashmiri breeds within the BP, P, and AP stages, 170/245/249 and 233/197/512 variants were detected within these genes. Among these variants in Assaf and Churra within the BP, P, and AP stages of lactation, 101/136/164 and 136/65/64 variants were novel, and 689/728/354 and 586/439/305 variants were previously annotated in SNPdb (version v97.0). In addition, within the Jersy and Kashmiri breeds within the BP, P, and AP stages of lactation, 30/22/37 and 43/22/61 variants were novel, and 140/223/212 and 190/175/451 variants were previously annotated in SNPdb (version v97.0) (Table 7).

Among all variants in milk fat-related genes, 179 were missense variants, out of which 170 and 9 were tolerate and deleterious variants, respectively. Among the missense variants detected in this study, one was in *ACACA*, seven were in *ACSL1*, five were in *LPL*, 12 were in *ACSS2*, 29 were in *XDH*, 13 were in *GPAM*, 59 were in *FASN*, one was in *VLDLR*, 24 were in *PLIN2*, and 28 were in the *BTN1A1* gene (Appendix A). A total of 60 novel missense variants were detected in milk fat genes. Among the missense variants detected in this study, 14 were in *XDH*, 25 were in *FASN*, two were in *VLDLR* and *GPAM*, nine were in *BTN1A1*, five were in *PLIN2*, one was in the *FABP3*, *ACSL1*, and *SCD* genes. Of these, 42 and 18 were tolerate and deleterious variants, respectively (Table 8).

### 3.5. Functional Enrichment Analysis

Assaf and Churra specific-SNPs in BP, P, and AP stages of lactation were located in 4530/5652/1984 and 3430/5284/1048 coding genes that were significantly (FDR < 0.01) enriched in 105/159/0 and 31/112/1 within the biological process (BP) category and the number of significant KEGG pathways included were 29/53/2 and 11/68/0, respectively. On the other hand, 138/1439/8198 and 640/2885/11,220 genes containing Jersy and Kashmiri specific SNPs in BP, P, and AP stages of lactation were significantly (FDR < 0.01) enriched in 0/18/529 and 0/205/91 of the biological process (BP) category. Due to of the large number of significant GO terms (biological process) and KEGG pathways, only the top 20 significant terms are displayed in the stages of BP, P, and AP for the Assaf, Churra, Jersy, and Kashmiri breeds.

Regarding significant biological processes in the BP stage of lactation, only Assaf and Churra sheep breeds showed significant terms. Five common enriched biological process terms, namely “organic substance metabolic process,” “macromolecule metabolic process,” “cellular metabolic process,” “primary metabolic process,” and “metabolic process,” were significant in the BP stage of lactation in Assaf and Churra sheep breeds (Figure 4). There was not any significant term in cow breeds.

The main and common significant biological process in the P stage of lactation in the Assaf, Churra, Jersy, and Kashmiri breeds include the “macromolecule metabolic process,” “cellular macromolecule metabolic process,” “cellular metabolic process,” and the “primary metabolic process” (Figure 5).

In the AP stage of lactation, the “organic substance metabolic process,” “macromolecule metabolic process,” “cellular macromolecule metabolic process,” “cellular metabolic process,” “primary metabolic process,” and “metabolic process” were common significant biological terms in Jersy and Kashmiri breeds and there were not any significant terms found in the sheep breeds (Figure 6).

The term “metabolic process” has several functions, and it is identified as a common term in three different stages of lactation among sheep and cow breeds.

Regarding significant KEGG pathways in the BP stage of lactation, only Assaf and Churra sheep breeds showed significant KEGG pathways. There were seven common enriched pathways, namely: “Protein processing in the endoplasmic reticulum,” “Fatty acid metabolism,” “Metabolic pathways,” “Phosphatidylinositol signaling system,” “Inositol phosphate metabolism,” “Adherens junction,” and the “FoxO signaling pathway” in the BP stage of lactation within the Assaf and Churra sheep breeds (Figure 7).

In the P stage of lactation, all breeds had significant KEGG pathways identified except for the Jersy cow breed. There were ten common pathways identified, including: “Regulation of actin cytoskeleton,” “FoxO signaling pathway,” “NF-kappa B signaling pathway,” “Adherens junction,” “Protein processing in endoplasmic reticulum,” “TNF signaling pathway,” “Fc gamma R-mediated phagocytosis,” “Phosphatidylinositol signaling system,” “T cell receptor signaling pathway,” and the “Osteoclast differentiation” pathway (Figure 8).

There was no common pathway in the AP stage among the Assaf, Jersy, and Kashmiri breeds. However, between the Kashmiri and Assaf breeds, the “Metabolic pathway” was a significant KEGG pathway (Figure 9). Also, the “TNF signaling pathway” was one of the most significant pathways in the Jersy cow breed (Figure 9).

## 4. Discussion

The SNP profile of two sheep and two cattle breeds was investigated to identify the potential contribution of genetic variants within the BP, P, and AP stages of the lactation process. To increase the accuracy of the variants called via RNA-Seq data, a strict filtering process was performed to prevent probable errors through computational analysis [11].

Approximately 10% and 20% of the recognized SNPs were novel in sheep and cow breeds, respectively. They have not been formerly annotated in the Ensembl ovine and bovine SNP databases. Our findings suggest that sheep and cow genetic diversity needs to be investigated further in more detail and demonstrate the lack of the existing annotations of these two species. This new process also accounts for non-annotated transcripts, which may code for a new protein. One of the main goals of the Functional Annotation of Animal Genomes (FAANG) project is to recognize these functional features in animals [24]. Furthermore, a higher quantity of SNPs recognized in different stages of lactation likely reflected a higher genetic variation in the lactation process. Here, we maximized reliable SNPs and concentrated on the function of the annotated SNPs [25].

In this research, only known breed-stage-specific SNPs were used for downstream analysis to provide a list of functional SNPs. The breed-stage-specific SNPs are presumed to be genetic variants that are significantly different between breeds and stages. However, the milk production phenotype is a complex attribute [1], and some known SNPs may also be associated with other traits that have functions other than being related to milk production. Therefore, to strengthen the results, we concentrated on the SNPs/genomic regions/genes positioned in QTL regions related to milk yield and milk composition traits. Considering the SNP distribution in both cow and sheep breeds, we showed that the variant density across the genome (Figure 3) had a non-uniform distribution.

To conclude the putative functions of the SNPs, their genomic location within the QTL analysis could be useful. A total of 278,110 breed-stages-specific SNPs were identified within the regions of sheep and cow QTLs responsible for milk yield and milk composition traits.

Within the coding region of the seven candidate genes that code for milk proteins, there were a large number of variants within the milk protein genes of the two sheep breeds that are related to the whey proteins group. However, in two cow breeds, a large number of milk protein variants belong to the casein cluster in all stages of lactation (Table 6).

All existing and novel mutations in the casein group within the two sheep breeds that were detected were found only in the *CSN3* (kappa casein) gene. Among the casein subtypes in sheep milk, the lowest percentage (percentage of total casein) is related to *CSN3* [26], probably due to these mutations observed in this study. In other words, these variants within this gene may be involved in decreasing the expression of this gene in casein subgroups. In the whey protein group, in sheep breeds, most mutations were observed in the alpha-lactalbumin (*LALBA*) gene, and the least mutations were observed in the beta-lactalbumin (*PAEP*) gene. Since *PAEP* has a higher percentage than *LALBA* in total sheep milk whey proteins [27], mutations observed in the *LALBA* gene may reduce the expression of this gene, and mutations in the *PAEP* gene may increase the expression of this gene in sheep milk during lactation. In addition, out of the total number of mutations observed in the whey protein group, two missense mutations (one in the *PAEP* gene and one mutation in the *LALBA* gene) were observed in the two sheep breeds, which are likely to play an important role in the decreased *LALBA* expression and increased *PAEP* expression observed in these breeds.

In both cow breeds, the highest number of mutations in the three stages of lactation was in the casein group, and the lowest was in the whey protein group. According to previous studies, the highest percentage of casein subtypes in cow milk belongs to the *CSN1S1* (αs1-casein) gene, and the lowest percentage belongs to the *CSN1S2* (αs2-casein) gene [26]. Therefore, it can be concluded that mutations observed in two bovine breeds in the *CSN1S1* gene increase the expression of this gene and vice versa, mutations in the *CSN1S2* gene reduce the expression of this gene.

Nine missense mutations were observed in casein-related genes in two bovine breeds in three different lactation stages. A mutation called rs43703016, which was identified as a deleterious mutation in all three BP-P-AP stages of both dairy cows, was identified in the *CSN3* gene. In the whey protein group, in both cow breeds, the lowest number of mutations was observed in the alpha-lactalbumin (*LALBA*) gene, and the greatest number of mutations were observed in the beta-lactalbumin (*PAEP*) gene. *PAEP* has a higher percentage of *LALBA* in total whey protein in cow’s milk [28], so mutations observed in the *LALBA* gene may decrease the expression of this gene, and mutations in the *PAEP* gene may increase the expression of this gene in cow’s milk, during lactation.

Three novel mutations in the casein group were observed in the *CSN1S2* gene in the AP stage of Jersey and Kashmiri, and one case in the AP stage of the Jersey breed. These mutations are reported for the first time in this study. Moreover, in the group related to whey proteins in dairy cows, three novel mutations were observed, one mutation was related to the AP stage of the Kashmiri breed, and the other two mutations were related to the AP stage of the Jersy breed. Most of the SNPs in milk protein genes, both in cow and sheep breeds, were identified within the intron region, and most of them were anticipated to be in the non-coding regions, which could explain why the highest genetic variation is observed in this region [8]. This finding is in agreement with previous variant calling research [8].

*CSN2* and *CSN3* genes are critical for the cheese-making process [29,30]. In sheep, the highest and lowest expression levels across the lactation phases are related to the functioning of *CSN2* and *CSN3*, respectively [31]. Milk casein micelles stabilization, which involves *CSN3* variants, has been related to protein content in sheep [32]. Micelles have different structures between sheep and cow milk, and their average diameter differs as well as their mineralization processes. The mineralization process in sheep milk is higher than in cow milk [33]. Micelles have different sizes, and in sheep milk, they are larger than what is found in cow milk [34]. On the other hand, there is a negative correlation between micelle size and casein concentration [34]. Micelle size also affects the rennet clotting time [34]. Sheep milk contains a high concentration of protein per casein unit, so it is an excellent material for cheese-making [35]. On the contrary, whey proteins are likely to impair cheese making, but they have a high level of essential amino acids (Tryptophan and Lysine) [36].

Previous studies have described significant associations among variants of the *PAEP* gene and protein percentage, fat percentage, clotting time, and curd firming time in milk [37]. Sheep milk is mainly used for the production of fine cheese varieties, yogurt, and whey cheeses [27]. The high levels of protein, fat, and calcium in casein result in an excellent matrix for cheese production [38].

Among the fatty acid synthesis-related genes, the *FASN* gene showed the largest number of SNPs. The high expression of the *FASN* gene is found in the mammary gland across the lactation process [13], which suggests that it plays a crucial role in fatty acid synthesis. The *ACACA* and *SCD* genes did not have any SNPs. Genes associated with the acetate triacylglycerol synthesis (*GPAM*) were the most highly expressed in the sheep mammary gland during lactation and fatty acid synthesis [13]. Thus, the related functional SNPs are of interest because they could influence milk composition and cheese-making.

The *XDH* gene, which is related to fatty acid metabolism, is responsible for milk fat globule secretion [39]. Hence, mutations in this gene could alter the mechanisms underlying lipid droplet secretion. *BTN1A1*, another gene that belongs to the fatty acid metabolism group, showed the highest expression during lactation in dairy cows [23], which is in agreement with the crucial role that it plays in milk fat secretion [40]. Thereby, these relevant functional SNPs found in the genes *XDH* and *BTN1A1* might affect the function of both proteins and, as a consequence, the lipid droplet formation process [8]. Whether this mutation can explain the higher milk fat contents of Churra sheep compared with Assaf sheep is of interest for further investigation.

Regarding the genes related to the fatty acid import into cells (*LPL* and *VLDLR*), there are not any deleterious SNPs detected. Two deleterious SNPs, namely rs380664726 and rs525585406, were found within the *PLIN2* gene in the BP/P/AP stages and AP stage of the Kashmiri breed, respectively. Adipophilin, which is encoded by *PLIN2*, is reported to play a role in the packaging of triglycerides for secretion as milk lipids in the mammary gland [41]. Moreover, in the formerly described sheep QTL for milk production traits, with the milk fat candidate genes considered here, we found a total of seven QTLs previously reported within the genomic regions of the *ACACA* and *DGTA1* genes in Altamurana, Gentile di Puglia, and Sarda sheep breeds [42]. Considering that the amount of fat yield increases lactation, it can be concluded that mutations observed in these genes at the beginning of the lactation process led to a decrease in the expression of genes involved in fat production. The identified and novel mutations in these genes during the peak and end of the lactation process are probably from an increase in the expression of these genes.

Among the GO analysis results, the metabolic process was the main function found across the lactation process in the sheep and cow breeds. Metabolic processes include a wide range of functions but typically transform small molecules (metabolites) and are involved in lipid, carbohydrate, and protein synthesis, as well as play a role in the degradation of these. The initial lactation period is usually considered a negative energy balance (NEB) process because the ingestion of nutrients and energy is not sufficient to produce the high energy demands of milk production. Therefore, balancing of energy demands and food intake for lactation in the early stages requires many metabolic processes. Based on previous research, the ontology of metabolic processes was considerably enriched for milk, fat, and protein yields in Nordic Red cattle [43]. Effects of an NEB in early and mid-lactation on performance, metabolic, and endocrine parameters have been reported previously [44], with markedly lower metabolic stress of NEB at the later lactation stage. Codrea et al. (2011) noted that the rate of recovery in milk yield due to a temporary nutritional shortage was unaffected by the stage of lactation [45]. Similarly, comparable losses in milk yield during short-term feed restrictions in early, mid-, and late lactation indicated that the metabolic responses due to feeding restrictions are dependent on milk yield [46]. However, the deviations of plasma metabolites from basal values were shown to be dependent upon the stage of lactation [46]. In particular, the early lactation period is known as when dairy cows are able to buffer against the metabolic load in diverse ways, and therefore this trait has been under artificial selective pressure (for example, otherwise there is an occurrence of metabolic disorders and loss in productive performance when failing to adapt).

In general, the “Fatty acid metabolism” pathway was one of the enriched pathways in the BP stage of lactation. During early lactation, fatty acid metabolism is ramped up and synchronized with this stage as an adaptation to address the NEB extreme energy demands [47]. The metabolic sub-pathways in relation to regulating lactation are closely integrated [48].

In the P and AP stages of lactation, some of the enriched pathways are related to immune system performance. For example, Fc gamma R-mediated phagocytosis, the TNF signaling pathway, the T cell receptor signaling pathway, and the NF-kappa B signaling pathway are connected with the metabolic functioning of the pathways affected by the different variants detected. Therefore, we suspect that improving or limiting the stress of the immune system can lead to the optimization of the lactation process, especially for the P and AP stages of lactation (milk production) [6].

## 5. Conclusions

In this study, we used RNA-Seq data to detect the variants involved in the BP, P, and AP stages of the lactation process in two sheep and two cow breeds. The comparison of the three different stages of the lactation process in both the sheep and cow species increased the probability of us identifying breed-stage-specific SNPs. The outcomes of this study focused on the identification of key SNPs and main pathways that are involved in the lactation process. Furthermore, the results of our study support the notion that the large number of breed-specific SNPs belonging to different genes and genomic regions identified are located in the genomic regions with known functions in milk yield and milk composition during the different stages of lactation. These variants were also mapped within QTL regions related to milk yield and composition traits. Findings of the present study also suggest that milk yield and milk composition in sheep and cow breeds at different stages of lactation can be related to known and novel variants of specific genes related to milk fat and protein synthesis. Our results pave the way for further research on determining the genetic basis of milk production and quality improvement for both domesticated sheep and cow breeds. The novel variants discovered here using RNA_seq data may be central and crucial when it comes time to design new SNP chips that are used as guides for selective breeding programs throughout the world. In summary, the novel variants detected in this study can be used for more precise genomic selection to substantially increase milk productivity in breeding programs.

## Figures and Tables

**Figure 1 animals-12-03592-f001:**
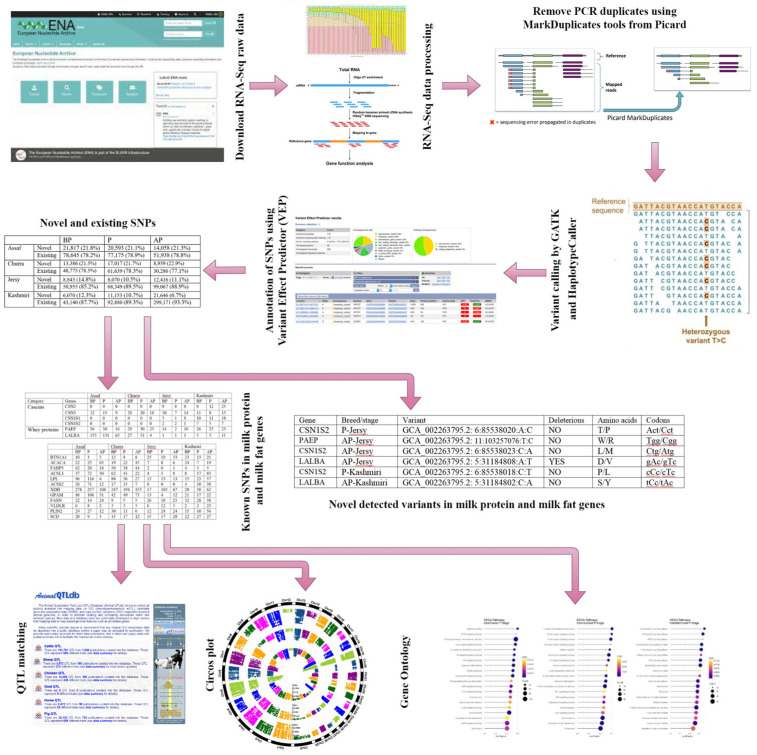
Graphical abstract of different bioinformatics analysis for SNP calling.

**Figure 2 animals-12-03592-f002:**
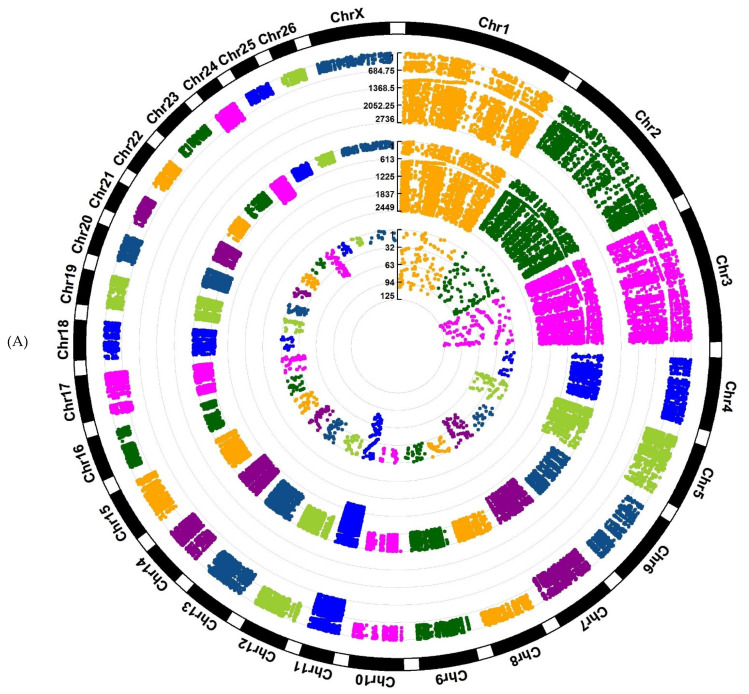
Circos plot of distribution and densities of sheep breed and stage-specific SNPs in (**A**) Assaf sheep breed and (**B**) Churra sheep breed. The external layer displays the chromosomes. The BP, P, and AP-specific SNPs are situated in the first, second, and third inner layers, respectively. Vertical lines in each layer represent the number of SNPs in that position.

**Figure 3 animals-12-03592-f003:**
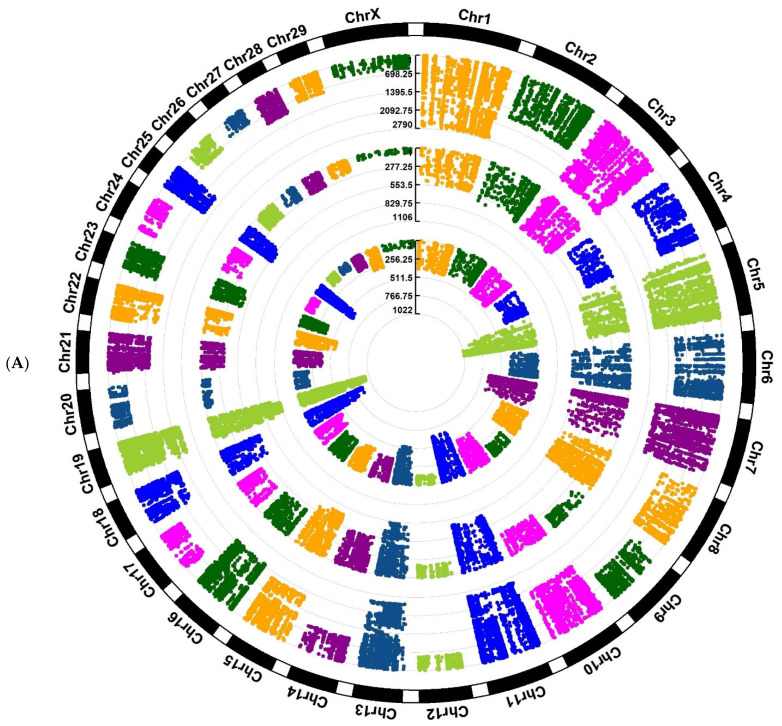
Circos plot of distribution and densities of cow breed and stage-specific SNPs in (**A**) Jersy cow breed and (**B**) Kashmiri cow breed. The external layer displays the chromosomes. The BP, P, and AP-specific SNPs are situated in the first, second, and third inner layers, respectively. Vertical lines in each layer represent the number of SNPs in that position.

**Figure 4 animals-12-03592-f004:**
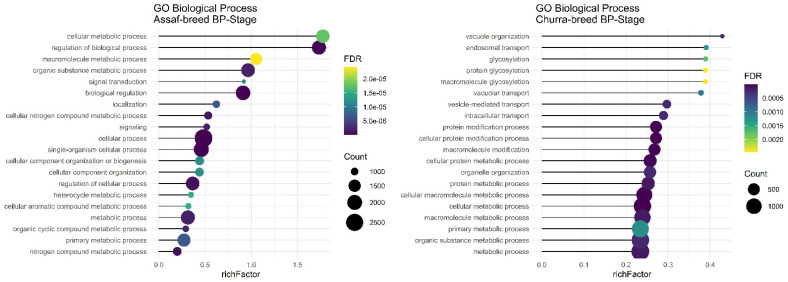
Functional enrichment analysis results of BP stage in Assaf and Churra sheep breeds. Since the significant biological process term is large, only 20 top significant terms are shown. The color and size of points indicate FDR and the number of genes associated with each term, respectively.

**Figure 5 animals-12-03592-f005:**
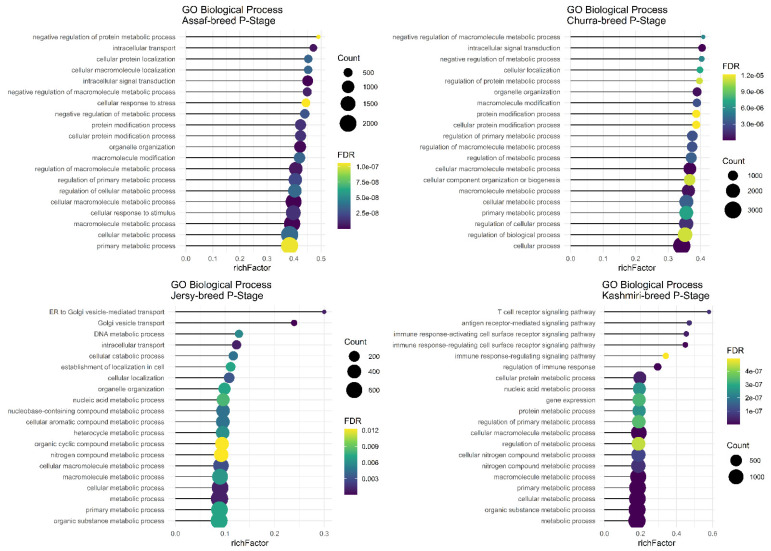
Functional enrichment analysis results of P stage in Assaf, Churra, Jersy, and Kashmiri breeds. Since the significant biological process term is large, only 20 top significant terms are shown. The color and size of points indicate FDR and the number of genes associated with each term, respectively.

**Figure 6 animals-12-03592-f006:**
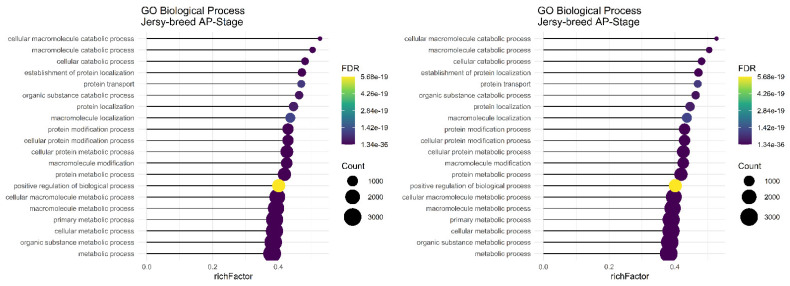
Functional enrichment analysis results of AP stage in Jersy and Kashmiri breeds. Since the significant biological process term is large, only 20 top significant terms are shown. The color and size of points indicate FDR and the number of genes associated with each term, respectively.

**Figure 7 animals-12-03592-f007:**
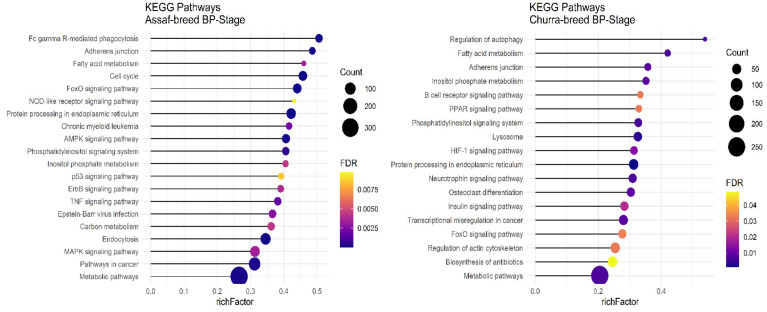
Functional enrichment analysis results of BP stage in Assaf and Churra sheep breeds. Since the significant KEGG pathways term is large, only 20 top significant terms are shown. The color and size of points indicate FDR and the number of genes associated with each term, respectively.

**Figure 8 animals-12-03592-f008:**
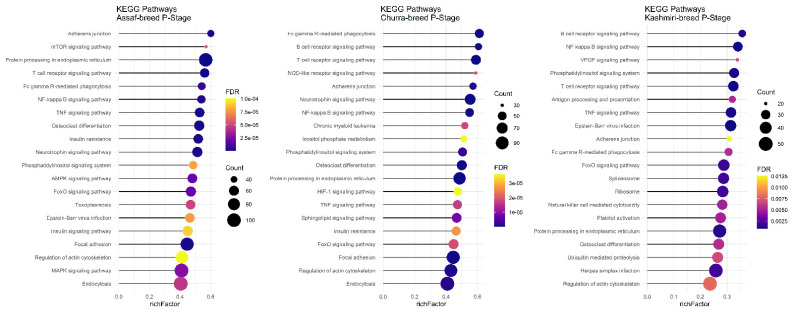
Functional enrichment analysis results of P stage in Assaf, Churra, and Kashmiri breeds. Since the significant KEGG pathways term is large, only 20 top significant terms are shown. The color and size of points indicate FDR and the number of genes associated with each term, respectively.

**Figure 9 animals-12-03592-f009:**
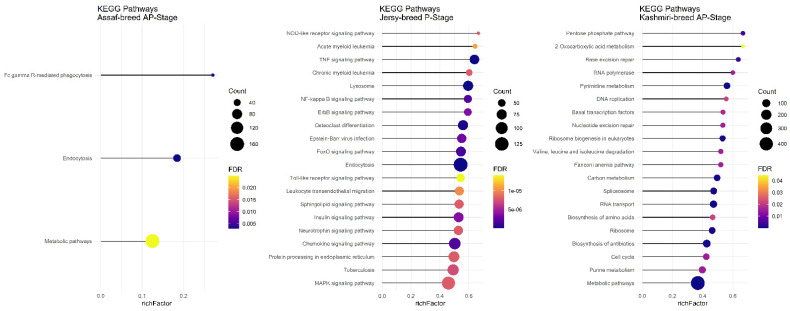
Functional enrichment analysis results of AP stage in Assaf, Jersy, and Kashmiri breeds. Since the significant KEGG pathways term is large, only 20 top significant terms are shown. The color and size of points indicate FDR and the number of genes associated with each term, respectively.

**Table 1 animals-12-03592-t001:** Dataset ID, species, and the number of specimens chosen for variant calling.

Accession ID	Species	Reference	Breed	No. of Samples			RNA Source
				BP	P	AP	
SRP125676	*Bos Taurus*	Bhat SA, et al. [5]	Jersy	3	2	3	MECs ^1^
SRP125676	*Bos Taurus*	Bhat SA, et al. [5]	Kashmiri	3	3	3	MECs ^1^
SRP065967	*Ovis aries*	Suárez-Vega, A., et al. [13]	Assaf	4	4	7	MFGs ^2^
SRP065967	*Ovis aries*	Suárez-Vega, A., et al. [13]	Churra	4	4	7	MFGs ^2^

^1^ MFGs = milk fat globules; ^2^ MECs = mammary epithelial cells.

**Table 2 animals-12-03592-t002:** The number of known and novel SNPs with corresponding percentages (in parenthesis) in the Ensemble database for sheep (Assaf and Churra) and cattle (Jersy and Kashmiri) breeds at different stages of lactation.

			Stage of Lactation 1		
Species	Breed	SNP	BP	P	AP
Sheep	Assaf	Novel	21,817 (21.8%)	20,593 (21.1%)	14,058 (21.3%)
		known	78,645 (78.2%)	77,175 (78.9%)	51,938 (78.8%)
	Churra	Novel	13,386 (21.5%)	17,017 (21.7%)	8959 (22.9%)
		known	48,775 (78.5%)	61,639 (78.3%)	30,286 (77.1%)
Cattle	Jersy	Novel	8843 (14.8%)	8070 (10.5%)	12,416 (11.1%)
		known	50,955 (85.2%)	68,349 (89.5%)	99,067 (88.9%)
	Kashmiri	Novel	6070 (12.3%)	11,153 (10.7%)	21,646 (6.7%)
		known	43,140 (87.7%)	92,880 (89.3%)	299,171 (93.3%)

^1^ Stages of lactation are before peak (BP), peak (P) and after peak (AP).

**Table 3 animals-12-03592-t003:** Characteristics of the identified breed and stage-specific SNPs in Assaf, Churra, Jersy, and Kashmiri breeds among three different stages of lactation.

Breeds	Assaf			Churra			Jersy			Kashmiri		
Stages	BP	P	AP	BP	P	AP	BP	P	AP	BP	P	AP
Total number	24,089	25,292	1150	11,564	25,138	2333	12,468	14,862	47,103	5785	13,903	215,074
**Effect by type (%)**												
3′-UTR	835	485	20	474	885	73	1159	1283	1342	771	1140	7080
5′-UTR	180	113	8	85	212	24	420	181	171	252	283	1019
Downstream gene	8670	5521	342	3834	8823	941	2419	2939	5972	1196	3014	23,171
Upstream gene	1341	1277	58	608	1225	119	1081	931	2935	659	900	6943
Intergenic	2332	2002	300	796	1581	263	740	1433	7636	431	690	17,422
Intron	5373	12,143	265	2987	6939	287	3724	5927	26,513	634	5301	145,453
non_coding_transcript_exon	389	215	12	201	397	64	81	40	88	59	43	208
splice_donor	19	19	0	10	16	8	20	7	28	17	14	20
splice_region	0	0	0	0	0	0	0	0	0	0	0	0
stop_gained	3	2	0	1	4	0	0	0	2	2	1	3
Missense	1329	911	46	677	1313	140	1024	709	826	536	905	4261
Synonymous	3579	2583	95	1883	3723	404	1787	1404	1577	1225	1604	9468
Others	39	21	4	8	20	10	13	8	13	3	8	26
**Effect by impact**												
High effect	58	39	0	18	35	17	33	12	43	22	22	45
Moderate effect	1329	911	49	677	1313	140	1024	709	826	536	905	4261
Low effect	3580	2585	96	1883	3727	405	1787	1407	1577	1225	1605	9471
Modifier	19,122	21,757	1005	8986	20,063	1771	9624	12,734	44,657	4002	11,371	201,297
**Deleterious**												
Deleterious SNPs	196	131	42	97	158	21	160	131	134	105	173	529

High effect; these SNPs have a disruptive impact on the protein. Moderate effect; these SNPs have a non-disruptive impact. Low effect; these SNPs have a harmless impact or change protein behavior. Modifier; these SNPs are non-coding.

**Table 4 animals-12-03592-t004:** The number of breed-stage-specific SNPs in milk QTL regions.

	BP	P	AP
Assaf	4710	8834	1892
Churra	3421	7537	714
Jersy	1325	1673	63,780
Kashmiri	672	4277	179,275

**Table 5 animals-12-03592-t005:** Annotated functionally relevant variants in genes coding for major milk proteins for sheep and cattle breeds at different stages of lactation.

		Assaf			Churra			Jersy			Kashmiri		
Category	Genes	BP	P	AP	BP	P	AP	BP	P	AP	BP	P	AP
Caseins	*CSN2*	0	0	0	0	0	0	9	8	8	8	12	25
	*CSN3*	12	13	9	21	20	18	10	7	14	11	8	15
	*CSN1S1*	0	0	0	0	0	0	3	1	8	10	11	18
	*CSN1S2*	0	0	0	0	0	0	2	2	1	7	5	7
Whey proteins	*PAEP*	36	30	16	29	30	23	14	2	10	26	23	23
	*LALBA*	155	131	65	27	51	4	1	1	1	5	5	11

**Table 6 animals-12-03592-t006:** Novel missense variants in major milk protein genes.

Gene	Breed/Stage	Variant	Deleterious	Amino Acids	Codons
*CSN1S2*	P-Jersy	GCA_002263795.2: 6:85538020:A:C	NO	T/P	Act/Cct
*PAEP*	AP-Jersy	GCA_002263795.2: 11:103257076:T:C	NO	W/R	Tgg/Cgg
*CSN1S2*	AP-Jersy	GCA_002263795.2: 6:85538023:C:A	NO	L/M	Ctg/Atg
*LALBA*	AP-Jersy	GCA_002263795.2: 5:31184808:A:T	YES	D/V	gAc/gTc
*CSN1S2*	P-Kashmiri	GCA_002263795.2: 6:85538018:C:T	NO	P/L	cCc/cTc
*LALBA*	AP-Kashmiri	GCA_002263795.2: 5:31184802:C:A	NO	S/Y	tCc/tAc

**Table 7 animals-12-03592-t007:** Functionally annotated relevant variants in the milk fat candidate genes considered in this study.

	Assaf			Churra			Jersy			Kashmiri		
	BP	P	AP	BP	P	AP	BP	P	AP	BP	P	AP
*BTN1A1*	10	3	5	12	8	8	25	19	33	23	13	21
*ACACA*	22	25	63	19	22	15	7	8	6	24	7	19
*FABP3*	62	20	18	50	38	14	2	0	1	4	3	5
*ACSL1*	37	72	50	62	41	22	4	5	8	8	17	83
*LPL*	96	116	6	80	36	27	13	13	13	15	23	57
*ACSS2*	26	71	12	17	15	7	0	0	0	4	10	38
*XDH*	278	257	108	247	194	155	17	103	67	20	18	62
*GPAM*	86	106	51	42	49	73	13	4	12	21	17	22
*FASN*	22	14	24	9	5	5	26	18	25	32	28	38
*VLDLR*	6	8	2	3	3	3	6	12	3	2	2	23
*PLIN2*	24	27	12	30	11	0	12	24	24	15	10	56
*SCD*	20	9	3	15	17	12	15	17	20	22	27	27

**Table 8 animals-12-03592-t008:** Novel missense variants in major milk fat genes.

Gene	Breed/Stage	Variant	Deleterious	Amino Acids	Codons
*XDH*	BP-Assaf	GCA_002742125.1:3: 98627777:T:C	NO	M/T	aTg/aCg
*XDH*	BP-Assaf	GCA_002742125.1:3: 98609204:A:T	NO	T/S	Acg/Tcg
*XDH*	BP-Assaf	GCA_002742125.1:3: 98619614:A:G	NO	I/V	Atc/Gtc
*XDH*	P-Churra	GCA_002742125.1:3:98592467:A:T	YES	E/V	gAg/gTg
*XDH*	P-Churra	GCA_002742125.1:3:98627777:T:C	NO	M/T	aTg/aCg
*XDH*	AP-Churra	GCA_002742125.1:3:98576977:G:T	YES	A/S	Gct/Tct
*XDH*	AP-Churra	GCA_002742125.1:3: 98627777:G:C	NO	M/T	aTg/aCg
*XDH*	P-Assaf	GCA_002742125.1:3: 98627777:T:C	NO	M/T	aTg/aCg
*XDH*	P-Assaf	GCA_002742125.1:3: 98627729:C:G	NO	T/S	aCc/aGc
*XDH*	P-Assaf	GCA_002742125.1:3: 98619614:A:G:	NO	I/V	Atc/Gtc
*XDH*	BP-Jersy	GCA_002263795.2: 11:14170914:G:T	NO	R/S	agG/agT
*XDH*	P-Jersy	GCA_002263795.2: 11:14201678:G:C	NO	E/Q	Gag/Cag
*XDH*	AP-Jersy	GCA_002263795.2: 11:14201678:G:C	NO	E/Q	Gag/Cag
*XDH*	AP-Jersy	GCA_002263795.2: 11:14219191:T:A	NO	V/E	gTg/gAg
*FASN*	BP-Assaf	GCA_002742125.1:11: 12323951:G:A	NO	S/N	aGc/aAc
*FASN*	P-Assaf	GCA_002742125.1:11: 12323951:G:A	NO	S/N	aGc/aAc
*FASN*	AP-Assaf	GCA_002742125.1:14: 64113338:T:C	NO	V/A	gTg/gCg
*FASN*	AP-Assaf	GCA_002742125.1:14: 64213154: T:C	NO	V/A	gTg/gCg
*FASN*	AP-Assaf	GCA_002742125.1:14: 64244350: T:C	NO	V/A	gTg/gCg
*FASN*	AP-Assaf	GCA_002742125.1:14: 64244524: T:C	NO	V/A	gTg/gCg
*FASN*	AP-Assaf	GCA_002742125.1:14: 66991561:G:A	NO	S/N	aGc/aAc
*FASN*	AP-Assaf	GCA_002742125.1:14: 66995097: G:A	NO	S/N	aGc/aAc
*FASN*	AP-Assaf	GCA_002742125.1:14: 66996605: G:A	NO	S/N	aGc/aAc
*FASN*	AP-Assaf	GCA_002742125.1:14: 67125964: G:A	NO	S/N	aGc/aAc
*FASN*	BP-Churra	GCA_002742125.1:14: 56312452:G:C	NO	G/A	gGc/gCc
*FASN*	BP-Churra	GCA_002742125.1:14: 56312518:G:C	NO	G/A	gGc/gCc
*FASN*	BP-Churra	GCA_002742125.1:14: 56312575:G:C	NO	G/A	gGc/gCc
*FASN*	BP-Churra	GCA_002742125.1:14: 56312591:G:C	NO	G/A	gGc/gCc
*FASN*	BP-Churra	GCA_002742125.1:14: 57589024:G:A	NO	S/N	aGc/aAc
*FASN*	BP-Churra	GCA_002742125.1:14: 57707905:G:A	NO	S/N	aGc/aAc
*FASN*	BP-Churra	GCA_002742125.1:14: 57744519:G:A	NO	S/N	aGc/aAc
*FASN*	BP-Churra	GCA_002742125.1:14: 57744737:G:A	NO	S/N	aGc/aAc
*FASN*	AP-Churra	GCA_002742125.1:11:12326870:T:G	YES	V/G	gTg/gGg
*FASN*	BP-Jersy	GCA_002263795.2: 19:50792414:A:T	YES	H/L	cAc/cTc
*FASN*	AP-Jersy	GCA_002263795.2: 19:50791738:G:T	YES	L/F	ttG/ttT
*FASN*	AP-Jersy	GCA_002263795.2: 19:50792940:G:T	YES	D/Y	Gat/Tat
*FASN*	AP-Kashmiri	GCA_002263795.2: 19:50789521:C:T	YES	A/V	gCc/gTc
*FASN*	AP-Kashmiri	GCA_002263795.2: 19:50791255:A:T	YES	I/F	Atc/Ttc
*FASN*	AP-Kashmiri	GCA_002263795.2: 19:50792935:C:A	NO	T/K	aCa/aAa
*PLIN2*	AP-Jersy	GCA_002263795.2: 8:25118513:C:T	YES	P/L	cCa/cTa
*PLIN2*	AP-Jersy	GCA_002263795.2: 8:25118516:A:C	YES	Q/P	cAa/cCa
*PLIN2*	AP-Jersy	GCA_002263795.2: 8:25121360:C:G	YES	L/V	Cta/Gta
*PLIN2*	AP-Jersy	GCA_002263795.2: 8:25123767:T:G	YES	C/W	tgT/tgG
*PLIN2*	P-Jersy	GCA_002263795.2: 8:25125321:C:A	YES	P/H	cCc/cAc
*BTN1A1*	BP-Jersy	GCA_002263795.2: 23:31586938:T:C	NO	F/L	Ttc/Ctc
*BTN1A1*	AP-Jersy	GCA_002263795.2: 23:31586938:T:C	NO	F/L	Ttc/Ctc
*BTN1A1*	BP-Kashmiri	GCA_002263795.2: 23:31586315:C:G	NO	D/E	gaC/gaG
*BTN1A1*	BP-Kashmiri	GCA_002263795.2: 23:31590953:A:T	NO	D/V	gAc/gTc
*BTN1A1*	P-Kashmiri	GCA_002263795.2: 23:31586315:C:G	NO	D/E	gaC/gaG
*BTN1A1*	AP-Kashmiri	GCA_002263795.2: 23:31586315:C:G	NO	D/E	gaC/gaG
*BTN1A1*	AP-Kashmiri	GCA_002263795.2: 23:31586913:A:C	YES	D/A	gAt/gCt
*BTN1A1*	AP-Kashmiri	GCA_002263795.2: 23:31586938:T:C	NO	F/L	Ttc/Ctc
*BTN1A1*	AP-Kashmiri	GCA_002263795.2: 23:31589254:G:T	YES	G/C	Ggc/Tgc
*SCD*	AP-Kashmiri	GCA_002263795.2: 26:21268941:C:A	YES	Q/K	Cag/Aag
*VLDLR*	BP-Assaf	GCA_002742125.1:2: 76235995:G:T	NO	R/L	cGc/cTc
*VLDLR*	P-Assaf	GCA_002742125.1:2: 76235995:G:T	NO	R/L	cGc/cTc
*FABP3*	P-Churra	GCA_002742125.1: 2:251668480:A:G	NO	I/V	Att/Gtt
*GPAM*	BP-Jersy	GCA_002263795.2: 26:32696082:G:T	YES	G/V	gGa/gTa
*GPAM*	BP-Kashmiri	GCA_002263795.2: 26:32725431:C:T	NO	T/M	aCg/aTg
*ACSL1*	BP-Jersy	GCA_002263795.2: 27:15189639:C:G	YES	L/V	Ctg/Gtg

## Data Availability

All used data in this study are available in https://www.ebi.ac.uk/ (accessed on 21 November 2018) database.

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
