# Peer review of "Intra- and Interspecies RNA-Seq Based Variants in the Lactation Process of Ruminants"

_animals, 2022, doi:10.3390/ani12243592_

Round 1

Reviewer 1 Report

In this study, the authors used publicly available RNA-seq data related to the lactation process of sheep and dairy cows to identify genetic variants involved in genes associated with different stages (before-peak, peak, and after-peak) of lactation from two sheep breeds and two dairy cow breeds. This study provides new insights into dairy and sheep breed selection and helps to design rational breeding programs.

There are some problems in the article, and the suggestions are as follows:

1. In the Abstract section, please refine the summary content.

2. In Results 3.1, the authors offer no new insights, which are recommended to be described in the Materials Methods section.

3. In Result 3.3, there are few QTL analysis results, so it is recommended to merge with the above results.

4. It is suggested to count the specific position (e.g. intron, exon region, etc.) of SNP in related genes.

5. In Figures 3 and 4, please indicate the breeds.

6. There are some verbal errors throughout the manuscript. For example, Line 20 “RNA_seq” and Line 27 “after-peak (BP)”.

Author Response

Response to Reviewer 1 Comments

Manuscript title:

Intra- and interspecies RNA-Seq based variants in the lactation process of ruminants

Thank you for giving us the opportunity to submit a revised draft of the manuscript “Intra- and interspecies RNA-Seq based variants in the lactation process of ruminants” for publication in the Journal of Animals. We appreciate the time and effort that you and the reviewers dedicated to providing feedback on our manuscript and are grateful for the insightful comments on and valuable improvements to our paper. We have incorporated most of the suggestions made by the reviewers. Those changes are highlighted within the manuscript. Please see below, for a point-by-point response to the reviewers’ comments and concerns.

Point 1: In the Abstract section, please refine the summary content.

Response 1:

Thank you for pointing this out. We checked the summary and refine it as follow:

RNA-Seq data provide new chance to find transcriptome variants. We used RNA-Seq data to detect the variants involved in the three different stages (before peak=BP, peak=P, and after peak lactation=AP) of the lactation process in two sheep and two cow breeds. Furthermore, several KEGG pathways and enriched gene ontologies associated with immune system activation and the metabolic process were demonstrated by analyzing the functional enrichment of the genes that were affected. Findings of the present study also suggest that milk yield and milk composition in both sheep and cow breeds at different stages of lactation can be related to known and novel variants of specific genes related to milk fat and protein synthesis. The results pave the way for further studies on determining the genetic basis of milk production. The novel variants discovered here using RNA_seq data may be central and crucial when it comes time to design new SNP chips that are used as guides for selective breeding programs.

Point 2: In Results 3.1, the authors offer no new insights, which are recommended to be described in the Materials Methods section.

Response 2:

We added the mapping information to the processing of RNA-Seq section.

We aligned the cleaned and trimmed reads of cow and sheep respectively on the Ovis aries (Oar_v4.0) and Bos taurus (ARS-UCD1.2) genomes, using Hisat2 (0.1.5-beta).

Point 3:  In Result 3.3, there are few QTL analysis results, so it is recommended to merge with the above results.

Response 3:

The QTL analysis section merged with variant calling and functional annotation.

Point 4: It is suggested to count the specific position (e.g. intron, exon region, etc.) of SNP in related genes.

Response 4:

Dear reviewer, in the Table 4, we put the most important information of the variants such as the count of variants in 3′-UTR, 5′-UTR, Downstream gene, Upstream gene, Intergenic, Intron, non_coding_transcript_exon, splice_region, stop_gained. All of this information were retrieved from VEP in the ENSEMBLE database.

Point 5:  In Figures 3 and 4, please indicate the breeds.

Response 5:

The name of the breeds was added to Figure 3 and 4.

Point 6: There are some verbal errors throughout the manuscript. For example, Line 20 “RNA_seq” and Line 27 “after-peak (BP)”.

Response 6:

Thank you for pointing this out. The manuscript was checked for all possibly errors and revised.

Reviewer 2 Report

Comments to the Authors,

The manuscript by Farhadian et al., investigated the intra- and interspecies RNA-Seq based variants in the lactation process of sheep and cow. This study reported that milk yield and milk composition in both sheep and cow breeds at different stages of lactation can be related to variants of specific genes related to milk fat and protein synthesis. However, the text is not sufficiently concentrated, and generally not easy to follow, major English changes required and could be a huge risk for showing a result with low confidence. Specific points are listed below.

Abstract:

-Line 27: the 2nd "BP" should be another abbreviation.

Introduction:

-Line 74: The abbreviation of “GWAS” should be added following “genome-wide association studies” rather than in Line 85.

-Line 89-90: The abbreviation of “QTL” should be added following “quantitative trait loci”.

Materials and Methods

-Line 114 Table 1: the RNA source of the animals are not from the same tissue could be a risk for the results. To my knowledge, there could be different gene expressions in different tissues even at the same stage. This risk could also be proved by Figure 2 that there is not even one common SNP between cow and sheep.

-Line 118-127: three different lactation stages BP, P, and AP are inaccurate, due to number of days in the same stage of lactation significantly affect gene expression in milk or mammary glands. I think it’s not suitable to compare cow and sheep in inaccurate stages in lactation with confusing definition.

-Line 118-127: the sampling is too limited and small sample size. The authors only used two independent studies from public datasets (SRP125676 and SRP065967) to compare the differences between two ruminant species. It is hard to validate the presented data and final conclusions.

Results

-Line 180: Please check “A of total”.

-Line 196: Please check “ovine”.

-Please concentrate the results of “3.6 Functional enrichment analysis”. Please re-write to reflect the key points of differences and common characteristics by bioinformatics analysis.

-Line 488: Please check whether is Table 5.

Conclusion

The final conclusions are not clear, for example, “…… in order to nourish an ever-growing world human population” is not correlative to this study.

Author Response

Response to Reviewer 2 Comments

Manuscript title:

Intra- and interspecies RNA-Seq based variants in the lactation process of ruminants

Thank you for giving us the opportunity to submit a revised draft of the manuscript “Intra- and interspecies RNA-Seq based variants in the lactation process of ruminants” for publication in the Journal of Animals. We appreciate the time and effort that you and the reviewers dedicated to providing feedback on our manuscript and are grateful for the insightful comments on and valuable improvements to our paper. We have incorporated most of the suggestions made by the reviewers. Those changes are highlighted within the manuscript. Please see below, for a point-by-point response to the reviewers’ comments and concerns.

Abstract:

Point 1: Line 27: the 2nd "BP" should be another abbreviation.

 Response 1:

Thank you for pointing this out. It was actually AP not BP and it changed to AP.

Introduction:

Point 2: Line 74: The abbreviation of “GWAS” should be added following “genome-wide association studies” rather than in Line 85.

Response 2: The given part checked and changed to “genome-wide association studies” based on dear reviewer comment.

Point 3: Line 89-90: The abbreviation of “QTL” should be added following “quantitative trait loci”.

 Response 3: Thank you for your comment. The full description of QTL added to the text.

Materials and Methods

Point 4: Line 114 Table 1: the RNA source of the animals are not from the same tissue could be a risk for the results. To my knowledge, there could be different gene expressions in different tissues even at the same stage. This risk could also be proved by Figure 2 that there is not even one common SNP between cow and sheep.

Response 4: Thank you for pointing this out. As the dear reviewer mentioned the RNA source of animals are not from the same tissue.

As you know, there are five different sources of RNA for investigation of milk production, namely mammary gland tissue (MGT), milk somatic cells (SC), laser microdissected mammary epithelial cell (LCMEC), milk fat globules (MFG), and antibody-captured milk mammary epithelial cells (mMEC). One of the simplest approaches to investigate the transcriptome association with milk appears to be the extraction of total RNA directly from SC or MFG released into milk during lactation. Canovat et al., (2014), indicated that the SC and MFG transcriptome are representative of MGT and LCMEC and can be used as effective and alternative samples to study mammary gland expression without the need to perform a tissue biopsy (Cánovas et al., 2014).

Cánovas, A., Rincón, G., Bevilacqua, C. et al. Comparison of five different RNA sources to examine the lactating bovine mammary gland transcriptome using RNA-Sequencing. Sci Rep 4, 5297 (2014). https://doi.org/10.1038/srep05297.

Point 5: Line 118-127: three different lactation stages BP, P, and AP are inaccurate, due to number of days in the same stage of lactation significantly affect gene expression in milk or mammary glands. I think it’s not suitable to compare cow and sheep in inaccurate stages in lactation with confusing definition.

Response 5: Thank you for pointing this out.

We did not divide the lactation period regarding the number of days. In a majority of mammalian species, the amount of milk production follows a curved pattern over the course of lactation. In early lactation, milk production peaks following an initial rise. After the peak yield, production gradually decreases until the end of lactation. But the length of three stages are different among species.

Farhadian et al., (2018), using 10 different attribute weighting models and counting the species as variable in addition to gene expression levels, showed that the developed meta-analysis signature of lactation is species-independent and is common among species.

Farhadian, M.; Rafat, S.A.; Hasanpur, K.; Ebrahimi, M.; Ebrahimie, E. Cross-species meta-analysis of transcriptomic data in combination with supervised machine learning models identifies the common gene signature of lactation process. Frontiers in genetics 2018, 9, 235.

Point 6: Line 118-127: the sampling is too limited and small sample size. The authors only used two independent studies from public datasets (SRP125676 and SRP065967) to compare the differences between two ruminant species. It is hard to validate the presented data and final conclusions.

Response 6: To compare the three stages of lactation, we selected the studies with samples related to three-stage. There were other studies but they didn’t cover the whole lactation period. So, we came to conclusion to use these datasets covering all lactation periods.

Also, the depth of these samples is high and it enables the results reliable. 

Results

Point 7: Line 180: Please check “A of total”.

Response 7: “A of total” changed to “Finally, a total of …”

Point 8: Line 196: Please check “ovine”.

Response 8: Thank you for your attention. It checked and changed to “bovine”.

Point 9: Please concentrate the results of “3.6 Functional enrichment analysis”. Please re-write to reflect the key points of differences and common characteristics by bioinformatics analysis.

Response 9: The most important and common feature in the functional enrichment analysis is that the different kind of metabolic process term enriched during three lactation process among two sheep and two cow breeds. So, this point added the functional enrichment analysis section in the results.

The term of "metabolic process" has several functions and it identified as a common term in three different stages of lactation among sheep and cow breeds.

Point 10: Line 488: Please check whether is Table 5.

Response 10: Thank you for pointing this out. It referrers to Table 6 and I changed it.

Conclusion

Point 11: The final conclusions are not clear, for example, “…… in order to nourish an ever-growing world human population” is not correlative to this study.

Response 11: In summary, the novel variants that have been detected in this study can be used for more precise genomic selection to substantially increase milk productivity in breeding programs.

Reviewer 3 Report

Reviewer comments on the manuscript “Investigation of intra- and interspecies RNA-Seq based variants in the lactation process of ruminants” by Mohammad Farhadian et al., submitted to the “Animals“ journal

The submitted work presents a thorough bioinformatic analysis of the RNA sequencing data obtained in previous accessions (from 2016 and 2019) in a public database ENA. Work contributed with identification of 6 novel missense variants in milk protein genes, which are otherwise well documented for variation. A similar result was obtained in milk fat metabolism-related genes, where a very high variability was demonstrated. Moreover, when combined with the positional information about relevant QTLs and the role in physiological pathways, the intended paper provides a powerful guide for subsequent association studies and breeding in two ruminant species, cattle and sheep. A top bioinformatic processing of data is obviously a based on the previous studies of the authors in this area.

Nevertheless, I have the following comments and questions to the authors:

1. The extent of the manuscript is not adequate. Most of journals adhere to the limit of 5000 words for the experimental work category. The current number of words exceeds 11000. Although the Diversity journal has no restrictions on the length of manuscripts, provided that the text is concise and comprehensive, the question is whether the current extent of the text is fully substantiated.

2. Analogically, the number of tables (9) and figures (10) in the text fairly exceed a usual amount of illustrating material in the category of original papers. For example, Table 3 (on the transversion to transition ratio in the found SNPs) is not very relevant to the main message of the paper.

3. The current abstract contains over 420 words instead of 200 recommended in the editorial instructions.

4. The original studies that generated the RNA-Seq data (Bhat et al., 2019; Suárez-Vega, A. et al., 2016, i.e., no 6 and 30 in the list), not speaking about the subsequent works of these groups, are cited only once, namely in the Table 1 in the Materials and Methods section. However, the results obtained in additional bioinformatic processing should be discussed in comparison to the originally reported interpretation. Therefore, the references to these two original papers should inevitably appear in the Discussion section.

5. SNP detection was based solely on one sequencing technology used in the works by Bhat et al. and Suárez-Vega, A. et al. Consequently, the identification of SNPs is affected by technical mistakes and some level of false SNPs can be assumed. It is not a problem for the already known SNPs, which are validated in the experimental material by a match with the database records. Novel SNPs are considered to be validated if they are confirmed using an independent technology, like designed genotyping reactions or resequencing with another technology. This problem is minimized in the latest sequencing technologies but in 2016 it was not definitely real. Do the authors have any estimate of the erroneously diagnosed SNPs, or at least some validation results?

This possibility is also supported by a comparatively high percentage of novel variants reported for the thoroughly explored genomes of cattle and sheep (20 and 10%, respectively, on line 461).

6. The used number of the replicates of the sequenced samples from each lactation stage was around 3, which is not too much (line 182). However, the authors should determine themselves during the bioinformatic processing whether this number of replicates is sufficient.

7. In the flowchart in Fig. 1, a continuation arrow from the “Novel detected variants” is missing.

8. “QTL matching” caption instead of “QTL analysis” might be used in this figure (Fig. 1) for a faster orientation of the reader.

9. In Fig. 2A, B and C, there is no need to use such complicated Venn diagrams since no SNPs are shared between species.

To increase the resolution in the printed version, please, consider simplification of the graphics or focusation at details.

10. The length of vertical (better: radial) lines in the Circos diagrams (in Figs. 3, 4) is evenly distributed along individual chromosomes. However, there are marked differences among the chromosomes. This phenomenon is difficult to explain. The graph would rather correspond to a chromosome-specific activation.

11. In Fig. 6, 7 and 8, the lettering is almost below the resolution level in the printed version. Moreover, the symbols are also close to the lower limit. Increasing the fonts and the figure modification might be considered. 

12. I suggest to separate the text blocks and paragraphs in Discussion with (numbered) subtitles.

The same might be useful in the Results section.

In some parts of the Discussion, simply the obtained results are listed, e.g., in the paragraphs starting on lines 511 and 520. These “result-like” parts should be minimized.

13. There is a potential for reducing the text in Conclusions, maybe by 50%.

14. There are enough references for an original paper, their number corresponds rather to a review (n = 58). Consequently, the paper might be unwieldy to read. Most of them are probably well substantiated, but a possibility of reduction might be beneficial for a manuscript as a whole.

The minor corrections or alternatives for the consideration of the authors are indicated in the attached pdf file of the manuscript (animals-1950177-peer-review-v1_unlocked+changes.pdf, appears as peer-review-23106016.v1.pdf).

Author Response

Response to Reviewer 3 Comments

Manuscript title:

Intra- and interspecies RNA-Seq based variants in the lactation process of ruminants

Thank you for giving us the opportunity to submit a revised draft of the manuscript “Intra- and interspecies RNA-Seq based variants in the lactation process of ruminants” for publication in the Journal of Animals. We appreciate the time and effort that you and the reviewers dedicated to providing feedback on our manuscript and are grateful for the insightful comments on and valuable improvements to our paper. We have incorporated most of the suggestions made by the reviewers. Those changes are highlighted within the manuscript. Please see below, for a point-by-point response to the reviewers’ comments and concerns.

Point 1: The extent of the manuscript is not adequate. Most of journals adhere to the limit of 5000 words for the experimental work category. The current number of words exceeds 11000. Although the Diversity journal has no restrictions on the length of manuscripts, provided that the text is concise and comprehensive, the question is whether the current extent of the text is fully substantiated.

Response 1: As the dear reviewer mentioned most of the journals has limitation in the word counts. I would like to mention that we try to summarize the text in all sections but since we investigate three time point in two different species, we had no choice to cover all aspect.

Point 2: Analogically, the number of tables (9) and figures (10) in the text fairly exceed a usual amount of illustrating material in the category of original papers. For example, Table 3 (on the transversion to transition ratio in the found SNPs) is not very relevant to the main message of the paper.

Response 2: The table 3 removed from the text.

Point 3: The current abstract contains over 420 words instead of 200 recommended in the editorial instructions.

Response 3: We summarized the abstract based the instruction word count.

Point 4: The original studies that generated the RNA-Seq data (Bhat et al., 2019; Suárez-Vega, A. et al., 2016, i.e., no 6 and 30 in the list), not speaking about the subsequent works of these groups, are cited only once, namely in the Table 1 in the Materials and Methods section. However, the results obtained in additional bioinformatic processing should be discussed in comparison to the originally reported interpretation. Therefore, the references to these two original papers should inevitably appear in the Discussion section.

Response 4: The original studies done differentially expressed genes in their works at different time points of lactation with different comparisons. Since in this study we performed variant calling and it’s not comparable with DEGs.

Point 5: SNP detection was based solely on one sequencing technology used in the works by Bhat et al. and Suárez-Vega, A. et al. Consequently, the identification of SNPs is affected by technical mistakes and some level of false SNPs can be assumed. It is not a problem for the already known SNPs, which are validated in the experimental material by a match with the database records. Novel SNPs are considered to be validated if they are confirmed using an independent technology, like designed genotyping reactions or resequencing with another technology. This problem is minimized in the latest sequencing technologies but in 2016 it was not definitely real. Do the authors have any estimate of the erroneously diagnosed SNPs, or at least some validation results?

This possibility is also supported by a comparatively high percentage of novel variants reported for the thoroughly explored genomes of cattle and sheep (20 and 10%, respectively, on line 461).

Response 5: Thank you for pointing this out. Detection of SNPs based one sequencing technology maybe given false SNPs and the experimental validation is needed especially for novel SNPs. That’s why and in order to minimize the false positive results in this study we combined the raw datasets of two cow together and two sheep together for variant calling.

Point 6: The used number of the replicates of the sequenced samples from each lactation stage was around 3, which is not too much (line 182). However, the authors should determine themselves during the bioinformatic processing whether this number of replicates is sufficient.

Response 6: Thank you for pointing this out. To compare the three stages of lactation, we selected the studies with samples related to three-stage. There were other studies but they didn’t cover the whole lactation period. That’s why we came to conclusion to use this dataset covering all lactation periods. Also, the depth of these samples is high and it enables the results reliable.

Point 7: In the flowchart in Fig. 1, a continuation arrow from the “Novel detected variants” is missing.

Response 7: In this research, only known breed-stage-specific SNPs were used for down-stream analysis to provide a list of functional SNPs. That’s why there is not any continuation arrow from the novel detected variants.

Point 7: “QTL matching” caption instead of “QTL analysis” might be used in this figure (Fig. 1) for a faster orientation of the reader.

Response 8: We changed the “QTL matching” with the “QTL analysis” in the Fig. 1.

Point 8: In Fig. 2A, B and C, there is no need to use such complicated Venn diagrams since no SNPs are shared between species.

To increase the resolution in the printed version, please, consider simplification of the graphics or focusation at details.

Response 8: Our goal is to compare detected SNPs in three different stages of lactation. Firstly, we instruct a Venn diagram for each step (BP for sheep; BP for cow; P for sheep; P for cow; AP for sheep; AP for cow). In that case the number of figures would increase and that’s why we decided to put the number of detected SNPs in each stage of both cow and sheep in one Venn diagram.

Point 9: The length of vertical (better: radial) lines in the Circos diagrams (in Figs. 3, 4) is evenly distributed along individual chromosomes. However, there are marked differences among the chromosomes. This phenomenon is difficult to explain. The graph would rather correspond to a chromosome-specific activation.

Response 9: Thank you for pointing this out. Vertical lines in each layer represent the number of SNPs in that position. It’s based on the default of the package and there is not any option in the packages to draw individual vertical line for every single chromosome.

Point 10: In Fig. 6, 7 and 8, the lettering is almost below the resolution level in the printed version. Moreover, the symbols are also close to the lower limit. Increasing the fonts and the figure modification might be considered. 

Response 10: We changed all the figures with new ones with high resolutions.

Point 11: I suggest to separate the text blocks and paragraphs in Discussion with (numbered) subtitles. The same might be useful in the Results section.

Response 11: Before submitting the manuscript, we set the discussion with numbered subtitles but in the word template that the Animal journal put in the website there were not any subtitled in the discussion section. That’s why we had to change based the instruction guide.

Point 12: In some parts of the Discussion, simply the obtained results are listed, e.g., in the paragraphs starting on lines 511 and 520. These “result-like” parts should be minimized.

Response 12: Thank you for your attention. I checked the discussion again and revised them.

Point 13: There is a potential for reducing the text in Conclusions, maybe by 50%.

Response 13: We tried our best to reduce the conclusion section without changing the main context.

Point 14: There are enough references for an original paper, their number corresponds rather to a review (n = 58). Consequently, the paper might be unwieldy to read. Most of them are probably well substantiated, but a possibility of reduction might be beneficial for a manuscript as a whole.

Response 14: I checked again the text and some of the references had removed.     

Point 15: The minor corrections or alternatives for the consideration of the authors are indicated in the attached pdf file of the manuscript (animals-1950177-peer-review-v1_unlocked+changes.pdf, appears as peer-review-23106016.v1.pdf).

Response 15: All of the recommended correction done in the main text.

Reviewer 4 Report

Generally, the manuscript is easy to read and has some novelty.  Some points the authors might need to clarify:

1.       It is not clear why did the authors choose two species, what are the biology of lactation for each species (cattle and sheep) and what could expect to be in common as well as be different, perhaps more introduction needed to be added.

2.       Table 1: How did the authors select these data, did the authors search for all available data sets?

3.       Table 1: Since the data from different sources, the expression of genes will not be the same. How did the authors correct for these differences in the background or confounding effects?

4.       During lactation, gene expression is strongly influenced by nutrition, did the authors can find any information about it?

5.       Since the authors reperformed the variant calling and DE analyses, how did these results differ from the previous publications? 

Author Response

Response to Reviewer 4 Comments

Manuscript title:

Intra- and interspecies RNA-Seq based variants in the lactation process of ruminants

Thank you for giving us the opportunity to submit a revised draft of the manuscript “Intra- and interspecies RNA-Seq based variants in the lactation process of ruminants” for publication in the Journal of Animals. We appreciate the time and effort that you and the reviewers dedicated to providing feedback on our manuscript and are grateful for the insightful comments on and valuable improvements to our paper. We have incorporated most of the suggestions made by the reviewers. Those changes are highlighted within the manuscript. Please see below, for a point-by-point response to the reviewers’ comments and concerns.

Comments and Suggestions for Authors

Generally, the manuscript is easy to read and has some novelty.  Some points the authors might need to clarify:

Point 1: It is not clear why did the authors choose two species, what are the biology of lactation for each species (cattle and sheep) and what could expect to be in common as well as be different, perhaps more introduction needed to be added.

Response 1: Thank you for pointing this out. Our goal in this manuscript is to investigate the milk production and milk composition. As you know, world milk production is almost entirely derived from cattle, buffaloes, goats, sheep and camel. On the other hand, for covering all lactation process (before peak, peak, and after peak lactation) we need the samples from all three stages. After searching in the database, we found some data relating sheep and cows.

Point 2: Table 1: How did the authors select these data, did the authors search for all available data sets?

Response 2:  Thank you for your comment. To compare the three stages of lactation, we had to select the studies with samples related to three-stage of lactation. There were other studies but they didn’t cover the whole lactation period. That’s why we came to conclusion to use these datasets which covering all lactation periods.

Point 3: Table 1: Since the data from different sources, the expression of genes will not be the same. How did the authors correct for these differences in the background or confounding effects?

Response 3: Thank you for pointing this out. As the dear reviewer mentioned the RNA source of animals are not from the same tissue.

As you know, there are five different sources of RNA for investigation of milk production, namely mammary gland tissue (MGT), milk somatic cells (SC), laser microdissected mammary epithelial cell (LCMEC), milk fat globules (MFG), and antibody-captured milk mammary epithelial cells (mMEC). One of the simplest approaches to investigate the transcriptome association with milk appears to be the extraction of total RNA directly from SC or MFG released into milk during lactation. Canovat et al., (2014), indicated that the SC and MFG transcriptome are representative of MGT and LCMEC and can be used as effective and alternative samples to study mammary gland expression without the need to perform a tissue biopsy (Cánovas et al., 2014).

Cánovas, A., Rincón, G., Bevilacqua, C. et al. Comparison of five different RNA sources to examine the lactating bovine mammary gland transcriptome using RNA-Sequencing. Sci Rep 4, 5297 (2014). https://doi.org/10.1038/srep05297.

Point 4: During lactation, gene expression is strongly influenced by nutrition, did the authors can find any information about it?

Response 4: Based on the supplementary information in the original papers, all animals were kept in free stall housing, fed with balanced and the same ration and had no water restrictions.

Point 5: Since the authors reperformed the variant calling and DE analyses, how did these results differ from the previous publications? 

Response 5: Dear reviewer, in this study we just performed variant calling in sheep and cow breeds separately without combining them. But in our previous publication (Farhadian et al., 2021) we integrated the sheep and cow sample using meta-analysis approach.

Farhadian, M.; Rafat, S.A.; Panahi, B.; Mayack, C. Weighted gene co-expression network analysis identifies modules and functionally enriched pathways in the lactation process. Scientific reports 2021, 11, 1-15.

Round 2

Reviewer 1 Report

The author has answered all the questions.

Reviewer 4 Report

Dear Authors, 

Thank you for responding to all my comments. I would suggest the authors add some texts in the manuscript from your responses to points 1, 2 3, and 4 in my comments. It is important for the readers to know more about the methods as well as why did the authors choose these data for analysis. 

I also suggest the authors might use the full names instead of using the appreciation BP, P, and AP in the text ( it is fine in the figure).

If these abbreviations are used in the tables, the authors might add a footnote for explaining what they are. Such as in table 6, the authors use P for both animal acid names, and for Peak; it is not suitable. Also, in the text, the authors wrote the full names of amino acids, I believe it is better to write them in full names in table 6 (easy for the reader to check).